# Reinforced Internal-External Knowledge Synergistic Reasoning for Efficient Adaptive Search Agent

## Abstract

Retrieval-augmented generation (RAG) is a common strategy to reduce hallucinations in Large Language Models (LLMs). While reinforcement learning (RL) can enable LLMs to act as search agents by activating retrieval capabilities, existing ones often underutilize their internal knowledge. This might lead to redundant retrievals, potential harmful knowledge conflicts, and increased inference latency. To address these limitations, an efficient adaptive search agent capable of discerning optimal retrieval timing and synergistically integrating parametric (internal) and retrieved (external) knowledge is in urgent need. This paper introduces the Reinforced **I**nternal-External **K**nowledge Synergistic R**E**asoning **A**gent (**IKEA**), which could indentify its own knowledge boundary and prioritize the utilization of internal knowledge, resorting to external search only when internal knowledge is deemed insufficient. This is achieved using a novel knowledge-boundary aware reward function and a knowledge-boundary aware training dataset. These are designed for internal-external knowledge synergy oriented RL, incentivizing the model to deliver accurate answers, minimize unnecessary retrievals, and encourage appropriate retrievals only when its own knowledge is lacking. Evaluations across multiple knowledge-intensive reasoning tasks demonstrate that IKEA significantly outperforms baseline methods, reduces retrieval frequency significantly, and exhibits robust generalization capabilities.

## 1 Introduction

The advancement of reinforcement learning with verifiable reward (RLVR) systems (Shao et al., 2024; Su et al., 2025) has significantly enhanced the reasoning capability of language models, like Deepseek-R1 (DeepSeek-AI et al., 2025). For knowledge-intensive tasks (Gao et al., 2024), R1-like models (Su et al., 2025) could activate their internal pre-trained knowledge through reasoning. However, constrained by the finite nature of pre-training corpora and the dynamic essence of world knowledge, they remain susceptible to hallucinations (Huang et al., 2025a). To address the knowledge deficiencies, current research typically empowers models to invoke search engines, essentially training them as search agents (Jin et al., 2025; Chen et al., 2025). With RL, these models progressively learn to decompose tasks and then retrieve relevant knowledge for each subtask to aid reasoning.

Despite this, the approach remains suboptimal for several reasons: **Firstly**, it primarily leverages the tool-calling and information extraction capabilities of LLMs, largely underutilizing its potential as an intrinsic knowledge base (i.e., LLM-as-KB (Heinzerling & Inui, 2021; Zheng et al., 2024)). This leads to substantial retrieval redundancy, as external searches are still performed even when the necessary information might already be implicitly encoded within the model parameters. **Secondly**, the redundant retrieval might introduce noise into the retrieved contents (Dong et al., 2025), potentially generating unnecessary knowledge conflicts (Fang et al., 2024). A common issue is erroneous retrieved knowledge overriding the accurate parametric knowledge (Xu et al., 2024b). **Thirdly**, since each search engine call interrupts the generation process of the LLM, an increase in the number of searches (caused by the aforementioned redundancy) will elevate inference latency (Yu et al., 2024). Thus, a critical research question emerges: ***How can we train an efficient adaptive search agent that comprehensively integrates both parametric (internal) and retrieved (external) knowledge?***

This paper argues that such an agent needs to posses the following key knowledge-related behaviors: (1) Self-knowledge Boundary Division (determine known/unknown): the ability to decompose a query into atomic queries and determine whether each sub-query falls within the knowledge boundary of the agent (Li et al., 2024a; Ren et al., 2024; Wen et al., 2024); (2) Internal Knowledge Recall (search in parameter): the ability to generate relevant background knowledge to assist in answering questions that fall within its knowledge boundary (Cheng et al., 2024; Mao et al., 2021); (3) External Knowledge Recall (search in corpus): the ability to generate effective search queries for questions outside its knowledge boundary and utilize search engines to acquire the desired knowledge (Zhao et al., 2024). In all, an efficient adaptive search agent needs to accurately determine whether to search in parameter or corpus, and should prioritize utilizing the knowledge embedded in its parameters, thereby minimizing redundant search for information that already exists within the model. Therefore, the retrieval timing becomes the core. Existing researches determine retrieval timing either via external indicators/classifiers (Jiang et al., 2023; Jeong et al., 2024), which often generalize poorly or require external tools, or through complex data engineering for imitation/preference learning (Yu et al., 2024; Guan et al., 2025; Wang et al., 2025a) to enable autonomous decision-making. However, how to imbue a model with the capacity to determine the optimal retrieval timing for adaptive retrieval via RL has not been fully investigated.

To address these issues and enable the model to exhibit the aforementioned behaviors, the paper proposes the Reinforced **I**nternal-External **K**nowledge Synergistic R**E**asoning **A**gent (**IKEA**), an efficient adaptive search agent powered by RL. First, we design the IKEA agent framework, which explicitly prompts the model to determine its knowledge boundary and prioritize the utilization of knowledge within its parameters. A search engine is invoked to retrieve external knowledge only when internal knowledge is deemed uncertain or insufficient. Next, we introduce two key components: a knowledge-boundary aware reward function and a corresponding knowledge-boundary aware training dataset for internal-external knowledge synergy oriented RL. The reward function incentivizes correct answers while minimizing unnecessary external knowledge retrieval for questions where the LLMs have possessed sufficient internal knowledge. Conversely, it encourages retrieval for questions beyond its knowledge boundary. In this way, the perception capability of self-knowledge in LLMs could be improved. Specifically, the training dataset, meticulously constructed, comprises a mixture of questions that the model is likely to answer through its internal knowledge and those requiring external knowledge. Actually, such a balanced dataset is crucial for the model training to adaptively and synergistically leverage both internal and external knowledge.

We conducted evaluations on multiple datasets involving both single-hop (Kwiatkowski et al., 2019; Mallen et al., 2023) and multi-hop (Yang et al., 2018; Ho et al., 2020) knowledge reasoning tasks. IKEA outperforms baselines across various datasets and demonstrates strong generalization capabilities on out-of-distribution datasets. Compared to naive reinforcement learning approaches (i.e. Search-R1) (Jin et al., 2025; Song et al., 2025; Chen et al., 2025), it can significantly reduce the number of retrievals while improving performance. This fully showcases the effectiveness and efficiency of our proposed method. The contribution of this paper are as follows:

- This paper addresses the limitations of current search agents, which often over-rely on external searches and underutilize their intrinsic knowledge, leading to retrieval redundancy.

- This paper proposes Reinforced Internal-External Knowledge Synergistic Reasoning Agent (IKEA), an efficient adaptive search agent via reinforcement learning, which could delineate the self-knowledge boundary and prioritize parametric knowledge before resorting to external retrieval.

- This paper presents a detailed analysis demonstrating that both knowledge-boundary aware reward design and training dataset construction are key to training efficient adaptive search agents.

## 2 PRELIMINARY

### 2.1 MULTI-TURN RLVR FOR LLM-AGENT

We consider an LLM agent $\pi$ that interacts with an environment $E$ over $N$ rounds to complete a task $t$. In each round $k$, the agent, based on the current state $s_k$, generates an action $a_k$. The environment responds with an observation $o_{k+1}$. The state $s_k$ is the history of all preceding tokens: $s_k = (t, a_0, o_1, \ldots, a_{k-1}, o_k)$. A full trajectory is denoted as $\tau = (t, a_0, o_1, \ldots, a_{N-1}, o_N, r)$, where $r$ is a final reward. RL aims to optimize the policy $\pi(a|s)$ to maximize this reward.

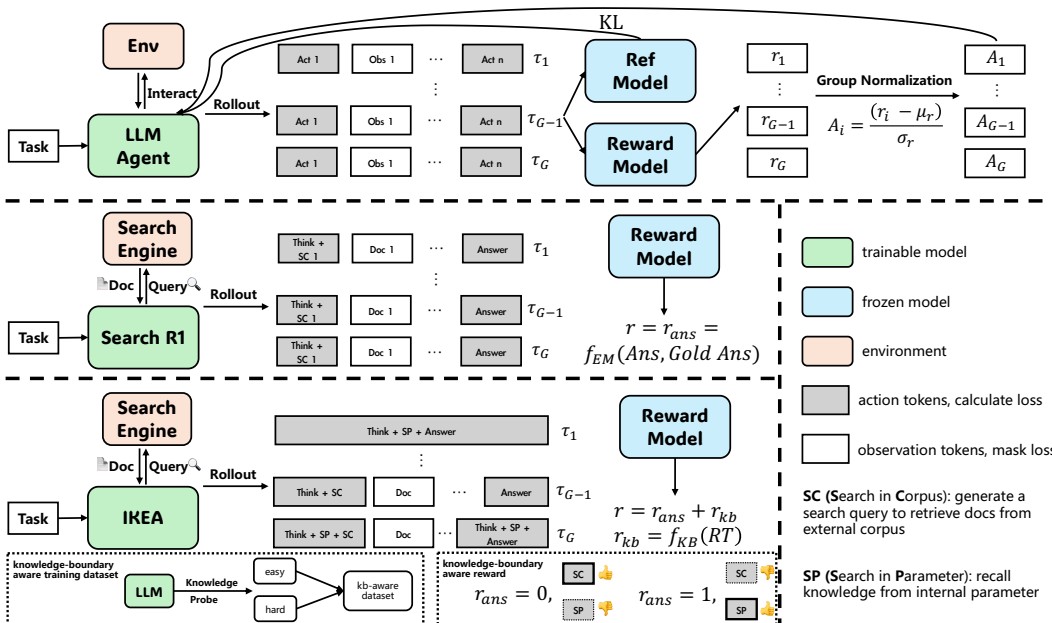

Figure 1: The top of the figure illustrates the training process for Multi-turn Reinforcement Learning with Verifiable Reward for LLM-Agent. In the middle is Search-R1, and at the very bottom is IKEA. Search-R1 and IKEA are special types of LLM-agents. We highlight the differences from the training of general LLM-agents, and to save space, we have omitted the common parts, such as the calculation of KL and Advantage.

While Proximal Policy Optimization (PPO) (Schulman et al., 2017) is a common practice, it requires training a separate, memory-intensive value model to estimate the advantage. To circumvent this, we adopt Group Relative Policy Optimization (GRPO) (Shao et al., 2024), a value-free algorithm. GRPO performs a group of $G$ rollouts per task and estimates the advantage for each trajectory $\tau_i$ by normalizing its reward $r_i$ relative to the group's statistics: $\hat{A}_{\tau_i} = (r_i - \mu_r)/\sigma_r$, where $\mu_r$ and $\sigma_r$ are the mean and standard deviation of rewards in the group. The GRPO loss function is:

$$\mathcal{L}^{\text{GRPO}}(\theta) = -\hat{\mathbb{E}}_{t \sim T, \tau_i \sim \pi_{old}(\tau|t)} \frac{1}{G} \sum_{i=1}^{G} \frac{1}{\sum_{k=0}^{N-1} |a_{i,k}|} \sum_{k=0}^{N-1} \sum_{\ell=1}^{|a_{i,k}|} \left[ \min \left( r_\theta \hat{A}_{\tau_i}, \text{clip}(r_\theta, 1 - \epsilon, 1 + \epsilon) \hat{A}_{\tau_i} \right) \right]$$

(1)

where $r_\theta = \frac{\pi_\theta(a_{i,k,l}|s_{i,k})}{\pi_{\theta_{old}}(a_{i,k,l}|s_{i,k})}$. The loss is computed only on the agent-generated action tokens.

## 2.2 AGENTIC SEARCH RL PARADIGM

We instantiate this RL framework for knowledge-intensive reasoning tasks, where the agent interacts with a search environment. The agent's actions $a_k$ are structured token sequences using special tags, as pioneered by recent works (Jin et al., 2025; Song et al., 2025; Chen et al., 2025). Each action must contain a reasoning step and an interaction step:

$$a_k = (\text{<THINK>} \ldots \text{</THINK>}, a_k^{\text{interact}})$$

The interaction part, $a_k^{\text{interact}}$, is either a search query or a final answer:

$$a_k^{\text{interact}} \in \{\text{<SEARCH>}[\text{query}]\text{</SEARCH>}, \text{<ANSWER>}[\text{final answer}]\text{</ANSWER>}\}$$

If a <SEARCH> action is generated, the environment returns retrieved documents as the observation $o_{k+1} = \text{<CONTEXT>}[\ldots]\text{</CONTEXT>}$. If an <ANSWER> action is generated, the episode terminates, and a final reward $r$ is calculated based on an Exact Match Reward Function. A group of such trajectories is then used to optimize the agent with the $\mathcal{L}^{\text{GRPO}}(\theta)$ loss function defined in Section 2.1. The content within the environment-provided <CONTEXT> tags is masked out during loss calculation.

## 3 IKEA: Reinforced Internal-External Knowledge Synergistic Reasoning Agent

An efficient adaptive search agent should possess the ability to delineate its own knowledge boundary and leverage internal parametric knowledge as much as possible within this boundary, while employing retrieval for knowledge outside this boundary. To this end, we propose the Reinforced Internal-External Knowledge Synergistic REasoning Agent (**IKEA**) as shown in the bottom of Figure 1. We design the system prompt (in Appendix B) to encourages the agent to first delineate its own knowledge boundary and prioritize its internal parametric knowledge. The fundamental challenge, however, is training the agent to learn how to make this crucial decision dynamically. We describe the design philosophy of the reward mechanism first. Then we elaborate the specific details for knowledge-boundary aware RL.

### 3.1 Design Philosophy: Knowledge Boundary Perception via Online Exploration

To overcome the redundant retrievals that plague existing models, our core design philosophy is to train an agent that can perceive its own **knowledge boundary**. We define this boundary as the distinction between the model's inter parametric knowledge and the external knowledge. Rather than using static rules, we empower the agent to learn this boundary dynamically through RL.

During the rollout phase for collecting trajectories, we can observe the agent's actions across two axes: whether it performs a search and whether it answers correctly, which defines four key behaviors:

* **Behavior 1 (No Search, Correct):** The ideal case where internal knowledge is sufficient.
* **Behavior 2 (No Search, Incorrect):** The worst case where the agent is unaware of knowledge gap.
* **Behavior 3 (Search, Correct):** A successful case where the agent actively fills knowledge gap.
* **Behavior 4 (Search, Incorrect):** A suboptimal case where an attempt to fill knowledge gap fails.

Our reward mechanism is meticulously designed to instill a preference for these behaviors in the order of **1 > 3 > 4 > 2**. This hierarchy teaches the agent to perceive its knowledge boundary through a series of crucial trade-offs.

**First**, the preference for **Behavior 1 over 3** is key to promoting efficiency and confidence. When both paths lead to a correct answer, we reward the agent more for relying on its internal knowledge. This penalizes *redundant* searches—those performed out of uncertainty rather than necessity—and encourages the model to trust its own parameters, thereby solidifying its perception of the internal knowledge boundary. **Furthermore**, within Behavior 3 itself, we can perform more fine-grained boundary learning. If multiple trajectories with varying numbers of searches all lead to a correct answer, we reward those with fewer searches more highly. This penalizes the *redundant* searches present in the longer trajectories, further training the agent to distinguish precisely which pieces of knowledge require external support. **Next**, the preference for **Behavior 4 over 2** is designed to discourage inaction when the agent has knowledge deficiency. While both scenarios result in an incorrect answer, Behavior 4 demonstrates that the agent was at least aware of a potential knowledge gap and attempted to explore. Rewarding this exploration, even if unsuccessful, trains the agent to avoid being "confidently wrong" and instead seek external help when its internal knowledge is insufficient. **Finally**, the clear separation between correct and incorrect outcomes (**Behaviors 1 & 3 > 4 & 2**) ensures that accuracy remains the primary objective. The agent learns that any path to a correct answer is fundamentally better than any path to an incorrect one.

This creates a dynamic tension that forces the agent to learn a true knowledge-boundary awareness. It must constantly balance the efficiency of using internal knowledge against the risk of being incorrect, and the necessity of exploration against the cost of search. This philosophy of cultivating a nuanced self-awareness is operationalized through the specific RL framework detailed next.

### 3.2 Knowledge-boundary Aware RL

**Reward Function.** Due to the probabilistic nature of LLMs, existing LLMs have a blurred perception of their self-knowledge boundaries. They cannot definitively distinguish which questions pertain to internal knowledge and which require external knowledge. As shown in the bottom of the Figure 1,

for the same task, $\tau_1$ only uses internal knowledge, $\tau_{G-1}$ only use external knowledge, and $\tau_G$ uses both internal and external knowledge. Consequently, agents may exhibit knowledge misidentification behaviors, leading to the generation of hallucinated answers for questions outside their knowledge boundaries, while utilizing redundant retrieval to for questions within their knowledge boundaries.

To address this, we design a knowledge-boundary aware reward composed of several components. First, the answer reward ($r_{ans}$) is 1 if the final answer matches the gold answer, and 0 otherwise. Second, the knowledge boundary reward ($r_{kb}$) is determined as follows: if $r_{ans} = 1$, $r_{kb}$ is a linear function increasing as the retrieval times ($RT$) decrease, ranging from 0 to $r_{kb+}$. If $r_{ans} = 0$, then $r_{kb} = 0$ when the number of retrievals is 0, and $r_{kb} = r_{kb-}$ (a small value) when the number of retrievals is greater than 0. Finally, for the format reward, if the generated trajectory violates format constraints of IKEA, the total reward is -1; otherwise, it is $r_{ans} + r_{kb}$.

The expression for the reward function is as follows:

$$R = \begin{cases} -1 & \text{if trajectory format is incorrect} \\ r_{\text{ans}} + r_{kb} & \text{if trajectory format is correct} \end{cases} \quad (2)$$

$$r_{\text{ans}} = \begin{cases} 1 & \text{ans == gold ans} \\ 0 & \text{ans != gold ans} \end{cases}, r_{kb} = \begin{cases} r_{kb+} \times \left(1 - \frac{RT}{RT_{max}}\right) & \text{if } r_{\text{ans}} = 1 \\ 0 & \text{if } r_{\text{ans}} = 0 \text{ and } RT = 0 \\ r_{kb-} & \text{if } r_{\text{ans}} = 0 \text{ and } RT > 0 \end{cases} \quad (3)$$

Here, $RT_{max}$ denotes the maximum number of retrievals, $r_{kb-}$ is a small value, $r_{kb+}$ is the maximum possible knowledge boundary reward. During exploration, when the agent obtains the correct answer ($r_{ans} = 1$), it may utilize internal or external knowledge. The reward $r_{kb+}$ is designed to incentivize the agent to minimize retrieval attempts, thereby favoring the use of internal knowledge. Conversely, when the agent fails to obtain the correct answer ($r_{ans} = 0$), indicating high uncertainty regarding relevant knowledge, the reward $r_{kb-}$ encourages reliance on external knowledge. To prevent the development of excessive retrieval behavior, we establish $r_{kb-} \ll r_{kb+}$.

**Dataset Construction.** We use In-context Learning with three Chain-of-Thought exemplars to probe the internal knowledge of the model. For each question, we sample the answer $N$ times. A question is labeled $Q_{easy}$ if the correct answer is obtained at least once, indicating the model likely possesses the relevant knowledge. Otherwise, it's labeled $Q_{hard}$.

If the training dataset exclusively contained data from $Q_{easy}$, the model would be more likely to utilize internal knowledge during rollout, and relying solely on internal knowledge would yield higher rewards than using retrieval. Consequently, after full training, the model would tend to avoid retrieval for any question. Conversely, if the training dataset only comprised $Q_{hard}$ questions, the model would be more inclined to use external retrieved knowledge during the rollout, and using the retriever would result in higher rewards than not using it. Thus, after full training, the model would tend to use retrieval exclusively for all questions.

To achieve a balanced use of internal and external knowledge, we construct the training dataset with a 1:1 ratio of $Q_{easy}$ and $Q_{hard}$ questions. This promotes adaptive retrieval and synergy between internal and external knowledge.

Finally, based on our carefully constructed reward function and the training dataset, we optimize the agent towards internal-external knowledge synergy using the Loss Function 1.

## 4 EXPERIMENT

### 4.1 SETTING

Test sets (easy and hard subsets) were constructed like the training set (Section 3.2), including two in-distribution and two out-of-distribution sets (details in Appendix C). We benchmarked our method against baselines (Appendix D) using various model sizes and types, with training specifics in Appendix E. Performance was evaluated by exact match (EM) and efficiency by the number of valid searches (RT) (Jin et al., 2025).

Table 1: Overall performance based on Qwen2.5-3B(-Instruct) and Qwen2.5-7B(-Instruct). Models with the "-Zero" suffix are trained from the Base model, otherwise trained from the Instruct model. EM is the exact match, and RT is the number of valid searches. *** results are reproduced using the checkpoint released by the original paper, the Search-R1-Zero-3B might suffer from over-optimization and it is hard to count the RT. †DeepRAG results are from its original paper; the values are EM/RT.

| Method | NQ | | | | PopQA | | | | HotpotQA | | | | 2Wiki | | | | Avg | |
|---|---|---|---|---|---|---|---|---|---|---|---|---|---|---|---|---|---|---|
| | Easy | | Hard | | Easy | | Hard | | Easy | | Hard | | Easy | | Hard | | | |
| | EM | RT | EM | RT | EM | RT | EM | RT | EM | RT | EM | RT | EM | RT | EM | RT | EM | RT |
| **Qwen2.5-3B** | | | | | | | | | | | | | | | | | | |
| *w/o parameter update* (re-implementation) | | | | | | | | | | | | | | | | | | |
| Direct | 36.33 | 0 | 3.91 | 0 | 56.05 | 0 | 2.54 | 0 | 50.39 | 0 | 1.56 | 0 | 50.98 | 0 | 11.72 | 0 | 26.69 | 0.00 |
| RAG | 59.77 | 1 | 30.47 | 1 | 68.16 | 1 | 31.64 | 1 | 54.30 | 1 | 13.87 | 1 | 40.04 | 1 | 12.70 | 1 | 38.87 | 1.00 |
| Iter-Retgen | 59.57 | 4 | 30.27 | 4 | 68.55 | 4 | 32.81 | 4 | 55.86 | 4 | 16.02 | 4 | 41.60 | 4 | 15.23 | 4 | 39.99 | 4.00 |
| IR-COT | 35.74 | 3.26 | 15.04 | 3.34 | 48.05 | 3.15 | 24.80 | 3.17 | 43.36 | 3.64 | 9.77 | 3.60 | 25.39 | 3.82 | 9.18 | 3.70 | 26.42 | 3.46 |
| FLARE | 34.57 | 0.21 | 4.49 | 0.41 | 52.15 | 0.18 | 4.49 | 0.52 | 48.44 | 0.16 | 1.76 | 0.61 | 50.00 | 0.03 | 11.13 | 0.32 | 25.88 | 0.31 |
| *Reinforcement learning* (re-implementation for search-r1) | | | | | | | | | | | | | | | | | | |
| R1-Zero | 62.34 | 0 | 10.55 | 0 | 72.66 | 0 | 3.71 | 0 | 59.57 | 0 | 4.49 | 0 | 57.22 | 0 | 13.28 | 0 | 35.48 | 0.00 |
| R1 | 59.77 | 0 | 7.23 | 0 | 70.11 | 0 | 3.13 | 0 | 58.01 | 0 | 3.71 | 0 | 57.81 | 0 | 13.67 | 0 | 34.18 | 0.00 |
| Search-R1-Zero*** | 66.60 | - | 28.51 | - | 77.73 | - | 27.93 | - | 64.45 | - | 13.67 | - | 52.54 | - | 13.48 | - | 43.11 | - |
| Search-R1 | 66.41 | 1.17 | 32.61 | 1.30 | 73.43 | 1.22 | 29.49 | 1.53 | 65.23 | 1.86 | 22.27 | 1.88 | 51.17 | 2.16 | 26.56 | 2.00 | 45.90 | 1.64 |
| **IKEA-Zero** | **71.29** | **1.00** | **34.18** | **1.00** | **78.90** | **1.00** | **35.94** | **1.02** | **68.94** | **1.05** | **21.09** | **1.14** | **54.69** | **1.19** | **23.63** | **1.39** | **48.58 (+5.47)** | **1.10** |
| **IKEA** | **72.46** | **1.00** | **31.44** | **1.02** | **79.69** | **1.00** | **33.59** | **1.02** | **69.92** | **1.04** | **20.11** | **1.13** | **59.37** | **1.15** | **20.70** | **1.21** | **48.41 (+2.51)** | **1.07 (-34.76%)** |
| **Qwen2.5-7B** | | | | | | | | | | | | | | | | | | |
| *w/o parameter update* (re-implementation) | | | | | | | | | | | | | | | | | | |
| Direct | 41.41 | 0 | 4.30 | 0 | 61.13 | 0 | 2.34 | 0 | 54.69 | 0 | 4.10 | 0 | 51.95 | 0 | 10.94 | 0 | 28.86 | 0.00 |
| RAG | 57.23 | 1 | 26.37 | 1 | 69.73 | 1 | 31.64 | 1 | 58.98 | 1 | 17.77 | 1 | 43.36 | 1 | 11.33 | 1 | 39.23 | 1.00 |
| Iter-Retgen | 58.79 | 4 | 26.95 | 4 | 70.90 | 4 | 31.25 | 4 | 61.52 | 4 | 19.73 | 4 | 43.36 | 4 | 14.45 | 4 | 40.87 | 4.00 |
| IR-COT | 40.04 | 2.59 | 14.26 | 2.68 | 58.98 | 2.48 | 25.78 | 2.56 | 46.09 | 3.09 | 14.26 | 2.94 | 17.58 | 3.18 | 12.50 | 3.07 | 28.69 | 2.82 |
| FLARE | 39.65 | 0.16 | 5.27 | 0.28 | 59.96 | 0.11 | 3.13 | 0.67 | 52.93 | 0.08 | 4.10 | 0.375 | 51.17 | 0.03 | 11.52 | 0.35 | 28.47 | 0.26 |
| *SFT/DPO* (results from the original paper, shown as EM/RT) | | | | | | | | | | | | | | | | | | |
| DeepRAG† | - | | | | 40.60/ | | | | 32.10/ | | | | 40.40/ | | | | - | - |
| *Reinforcement learning* | | | | | | | | | | | | | | | | | | |
| R1-Zero | 66.80 | 0 | 15.23 | 0 | 72.65 | 0 | 6.25 | 0 | 64.65 | 0 | 5.66 | 0 | 53.32 | 0 | 18.16 | 0 | 37.84 | 0.00 |
| R1 | 62.50 | 0 | 14.06 | 0 | 73.04 | 0 | 5.27 | 0 | 64.06 | 0 | 5.47 | 0 | 57.23 | 0 | 14.45 | 0 | 37.01 | 0.00 |
| Search-R1-Zero | 68.55 | 1.19 | 35.55 | 1.34 | 76.37 | 1.16 | 33.59 | 1.30 | 69.73 | 1.78 | 25.78 | 1.77 | 46.68 | 2.38 | 26.56 | 2.13 | 47.85 | 1.63 |
| Search-R1 | 65.63 | 1.34 | 33.40 | 1.51 | 78.13 | 1.24 | 32.62 | 1.51 | 68.17 | 2.00 | 24.02 | 2.07 | 35.35 | 2.67 | 22.66 | 2.47 | 45.00 | 1.85 |
| **IKEA-Zero** | **74.80** | **1.00** | **37.89** | **1.00** | **80.47** | **1.00** | **33.20** | **1.00** | **74.22** | **1.01** | **23.43** | **1.08** | **57.42** | **1.03** | **27.34** | **1.23** | **51.10 (+3.25)** | **1.04 (-36.20%)** |
| **IKEA** | **74.61** | **0.59** | **32.23** | **0.89** | **80.08** | **0.56** | **31.84** | **1.09** | **71.88** | **0.60** | **26.56** | **1.20** | **54.49** | **0.93** | **28.71** | **1.38** | **50.05 (+5.05)** | **0.91 (-50.81%)** |

(a) Num of Valid Search    (b) Response Length    (c) Training Rewards

Figure 2: The training log of IKEA-3B-Zero, IKEA-3B, IKEA-7B-Zero and IKEA-7B. We show the curve of number of valid searches, response length and trainign rewards.

## 4.2 OVERALL RESULTS

Experimental results are presented in Table 1, with corresponding training dynamics illustrated in Figure 2. A detailed analysis was conducted to demonstrate the advantages of the proposed method and provide insights for future research.

**Baselines without parameter updates struggle to effectively synergize internal and external knowledge.** "Direct" (internal knowledge) is outperformed by retrieval-based methods like "RAG" and "Iter-Retgen" (Shao et al., 2023) on "Hard" tasks, revealing LLMs' internal knowledge gaps. However, constant retrieval introduces conflicts and latency. Adaptive methods like IR-COT (Trivedi et al., 2023) and FLARE (Jiang et al., 2023) fail to solve this; IR-COT improves "Hard" performance at the cost of "Easy" tasks, while FLARE's token-probability trigger is ineffective, leading to minimal retrieval. This shows that un-finetuned models cannot autonomously and effectively determine when to synergistically leverage internal and external knowledge.

**Reinforcement learning baselines can effectively activate either internal or external knowledge, but not both synergistically.** R1, an RL-based method focusing on internal knowledge, improves performance on "Easy" subsets but not on "Hard" ones. Conversely, Search-R1 (Jin et al., 2025) uses RL to generate search queries for external knowledge, outperforming other retrieval methods with fewer calls. While both demonstrate RL's ability to enhance the use of internal or external knowledge in isolation, neither method effectively integrates the two sources.

**IKEA can adaptively combine internal and external knowledge for synergistic knowledge reasoning.** Through RL with a knowledge-boundary aware reward, IKEA learns to use internal knowledge when sufficient and retrieve external knowledge only when necessary. Compared to R1, IKEA improves performance by over 10%, primarily on "Hard" subsets. Compared to Search-R1, it achieves superior performance with significantly fewer retrievals. This shows IKEA learns to delineate its own knowledge boundaries, leveraging parametric knowledge by default and external knowledge selectively, which mitigates knowledge conflicts, improves efficiency, and generalizes well to out-of-distribution datasets.

**The IKEA training method is effective across models of different sizes and types.** As shown in Figure 2, IKEA models initialized from instruction-tuned LMs and IKEA-Zero models from base LMs converge to similar high rewards, demonstrating that RL can instill this synergistic reasoning capability from scratch. Larger models (7B vs. 3B) learn faster and achieve better results. The training logs show a pattern where retrieval counts first increase (exploration) and then decrease (exploitation), indicating that the model learns to eliminate retrieval redundancy over time.

### 4.3 ERROR ANALYSIS

A key motivation for IKEA is to mitigate errors caused by the flawed retrieval strategies of existing models. To analyze the effectiveness of IKEA in this regard, we categorize these common failures into two types that often stem from redundant retrieval:

**Invalid Retrieval:** This occurs due to insufficient internal knowledge, and the subsequent external search also fails to provide relevant information, leading to an incorrect final answer. **Conflict Retrieval:** This occurs when the model's internal parametric knowledge is actually correct, but the external search retrieves conflicting or misleading information that overrides this internal knowledge, also resulting in an incorrect answer.

To quantify performance against these specific failure modes, we created approximate error sets by analyzing the performance of the 7B-level R1 (internal knowledge only) and Search-R1 (external knowledge only) models on the HotpotQA dataset. Cases where both models failed are labeled as "Invalid Retrieval," while cases where R1 succeeded but Search-R1 failed are labeled as "Conflict Retrieval." As shown in Table 2, IKEA demonstrates a substantial ability to resolve these challenging cases.

Table 2: IKEA-7B Performance on Redundant Retrieval Error Types.

| Partition | Count | IKEA Correct | IKEA Correct Ratio |
|---|---|---|---|
| Invalid Retrieval | 452 | 55 | 0.12 |
| Conflict Retrieval | 90 | 57 | 0.63 |

IKEA's success in these scenarios stems directly from its core design philosophy of knowledge boundary perception: **Against Invalid Retrieval:** IKEA learns to generate more precise and effective search queries through reinforcement learning. By improving the quality of its exploration, IKEA can retrieve relevant information where Search-R1's more generic queries fail, thus successfully bridging the knowledge gap. **Against Conflict Retrieval:** Because IKEA's reward function encourages it to trust its internal knowledge when sufficient (Behavior 1), it learns to *avoid* performing redundant searches for topics it already knows well. This fundamental avoidance of unnecessary retrieval is its primary defense, as it prevents the model from ever being exposed to the potentially conflicting external information that misleads Search-R1.

The case studies shown in the Appendix F provide qualitative examples of how IKEA achieves this.

## 5 ABLATION STUDY

We conducted ablation studies based on Qwen2.5-3B-Instruct, which fully validated the effectiveness of the proposed method.

### 5.1 THE EFFECTS OF REWARD DESIGN

We present the training process using different rewards in Figure 3 and the final test results in Table 3. Without the knowledge boundary aware reward ("$w/o\ r_{kb}$"), both effective retrievals and response length show a consistent upward trend, significantly surpassing models with the original reward.

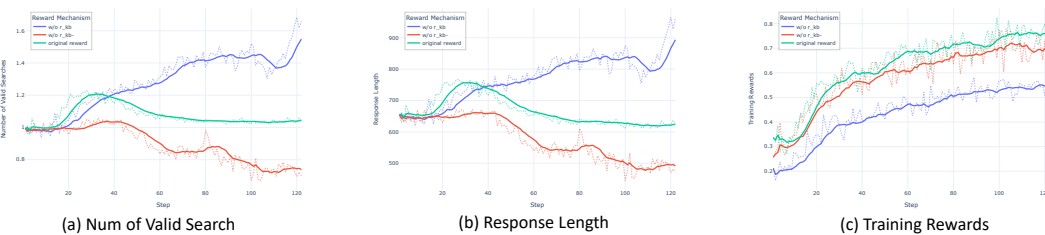

(a) Num of Valid Search      (b) Response Length      (c) Training Rewards

Figure 3: The training logs of different reward design. We show the curve of number of valid searches, response length and trainign rewards.

Table 3: The ablation results of reward design.

| Method | NQ | | PopQA | | HotpotQA | | 2Wiki | | Avg |
|---|---|---|---|---|---|---|---|---|---|
| | Easy | Hard | Easy | Hard | Easy | Hard | Easy | Hard | |
| IKEA (EM) | 72.46 | 31.44 | 79.69 | 33.59 | 69.92 | 20.11 | 59.37 | 20.70 | 48.41 |
| RT | 1.00 | 1.02 | 1.00 | 1.02 | 1.04 | 1.13 | 1.15 | 1.21 | 1.07 |
| IKEA w/o $r_{kb-}$ (EM) | 66.01 | 28.91 | 74.61 | 32.42 | 66.99 | 20.90 | 55.27 | 0.21 | 43.17 |
| RT | 0.48 | 0.68 | 0.53 | 1.00 | 0.58 | 1.08 | 0.64 | 1.11 | 0.89 |
| IKEA w/o $r_{kb}$ (EM) | 71.09 | 34.57 | 76.37 | 32.23 | 70.12 | 25.59 | 53.32 | 25.20 | 48.56 |
| RT | 1.40 | 1.54 | 1.35 | 1.63 | 1.94 | 2.12 | 2.40 | 2.48 | 1.86 |

This is because early in training, retrieval is more frequently rewarded than relying on parametric knowledge, leading to gradient updates that suppress the latter. Consequently, the model develops a bias for "retrieval > no retrieval", eventually maximizing reliance on retrieved knowledge, akin to the Search-R1 strategy. For the "$w/o\ r_{kb}$-" case (excluding the negative component of the knowledge boundary aware reward), retrieval count and response length are significantly less than the original reward. Because the positive reward component ($r_{kb}+$) encourages greater reliance on internal knowledge. This leads to incorrect generalization, where the model increasingly defaults to the R1 strategy even for questions requiring external knowledge. Final results show that IKEA "$w/o\ r_{kb}$" achieves a similar EM score but with significantly more retrievals. Conversely, IKEA "$w/o\ r_{kb-}$" exhibits considerably degraded performance alongside a substantial decrease in retrievals. Therefore, we conclude that an effective knowledge boundary aware reward function must appropriately balance internal and external knowledge utilization to achieve their synergistic application.

## 5.2 THE EFFECTS OF DATASET DIFFICULTY

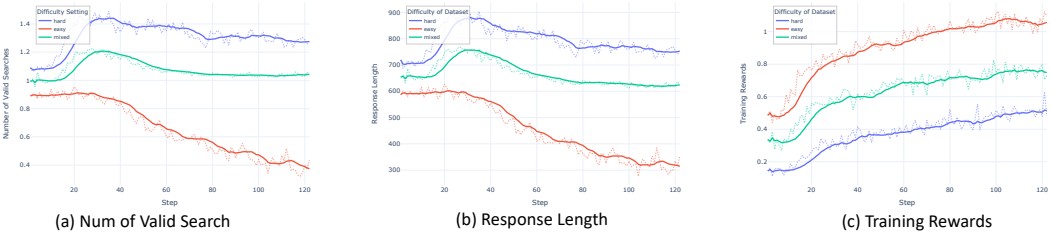

(a) Num of Valid Search      (b) Response Length      (c) Training Rewards

Figure 4: The training logs of different the difficulty of training datasets. We show the curve of number of valid searches, response length and trainign rewards.

We illustrate the training processes using different datasets of varying difficulty in Figure 4 and present the final test results in Table 4. Training on datasets of varying difficulty (easy, mixed, hard) revealed a consistent trend during training: Hard > Mixed > Easy for both effective number of searches and response length. This is because the model uses parametric knowledge for problems within its knowledge boundary and retrieval knowledge for those beyond it. Training on the Easy dataset showed a continuous decrease in retrieval attempts and response length, indicating that models

Table 4: The ablation results of the difficulty of the training datasets.

| Method | NQ | | PopQA | | HotpotQA | | 2Wiki | | Avg |
|---|---|---|---|---|---|---|---|---|---|
| | Easy | Hard | Easy | Hard | Easy | Hard | Easy | Hard | |
| IKEA (EM) | 72.46 | 31.44 | 79.69 | 33.59 | 69.92 | 20.11 | 59.37 | 20.70 | 48.41 |
| RT | 1.00 | 1.02 | 1.00 | 1.02 | 1.04 | 1.13 | 1.15 | 1.21 | 1.07 |
| IKEA w/ easy (EM) | 66.99 | 21.88 | 76.17 | 25.59 | 66.70 | 15.43 | 56.45 | 16.80 | 43.25 |
| RT | 0.28 | 0.54 | 0.29 | 0.80 | 0.34 | 0.84 | 0.16 | 0.70 | 0.49 |
| IKEA w/ hard (EM) | 66.02 | 33.98 | 75.39 | 0.35 | 64.65 | 25.00 | 46.09 | 23.63 | 41.89 |
| RT | 1.03 | 1.07 | 1.05 | 1.11 | 1.46 | 1.59 | 2.08 | 2.10 | 1.44 |

converge to behaviors characteristic of the training data's difficulty. On the test set, both easy and hard variants of the IKEA model showed substantially lower Exact Match (EM) scores compared to the original. Retrieval attempts dropped significantly for the easy variant and increased substantially for the hard variant. This highlights that disproportionately favoring one type of knowledge hinders full performance, underscoring the importance of synergistically using both internal (parametric) and external (retrieval-based) knowledge for effective reasoning.

## 6 RELATED WORK

**RL for LLM-based Agent**    Reinforcement Learning (RL) (Wang et al., 2025b) is a crucial technique for post-training Large Language Models (LLMs), enabling the alignment of pre-trained models' values (Ouyang et al., 2022) and enhancing their capabilities in specific downstream tasks (Goldie et al., 2025). The community has developed various distinctive RL algorithms, such as PPO (Schulman et al., 2017), DPO (Rafailov et al., 2023), RLOO (Ahmadian et al., 2024), ReMax (Li et al., 2024b), and GRPO (Shao et al., 2024). Building upon this, by constructing different environments and reward functions, LLMs can be trained into intelligent agents capable of autonomous decision-making and interaction with the environment (Huang et al., 2025b). A typical application in this area is the Search Agent (Jin et al., 2025; Chen et al., 2025; Song et al., 2025), which interacts with search engines to continuously acquire knowledge from the environment and perform reasoning.

**The Knowledge Boundary of LLM**    Large Language Models possess parametric knowledge (Zheng et al., 2024) and can access external knowledge. The concept of *Knowledge Boundary* (Li et al., 2024a; Xu et al., 2024a) distinguishes between these. This boundary is probed using template-based methods (evaluating responses to specific prompts (Petroni et al., 2019)) or internal state-based methods (classifying based on model features like hidden states (Chen et al., 2024) or SAEs (Zhao et al., 2025)). Understanding this boundary is crucial for Retrieval Augmented Generation (RAG) models (Ren et al., 2024) to adapt their behavior to different questions and avoid hallucinations.

## 7 CONCLUSION AND LIMITATIONS

This paper introduced the Reinforced Internal-External Knowledge Synergistic Reasoning Agent (IKEA), an innovative approach to developing efficient and adaptive search agents. IKEA addresses critical limitations in existing RL-based search agents, namely the underutilization of internal knowledge, which can lead to redundant retrievals, potential knowledge conflicts, and increased inference latency. The core of IKEA lies in its ability to discern its own knowledge boundary, prioritizing the use of its internal parametric knowledge and resorting to external search only when the internal knowledge is deemed insufficient or uncertain. This is achieved through a novel knowledge-boundary aware reward function and a meticulously constructed knowledge-boundary aware training dataset. This approach significantly enhances reasoning efficiency and accuracy on knowledge-intensive tasks. Despite these achievements, IKEA's reliance on specific dataset construction and model probing for knowledge boundary awareness may limit its universal applicability, the reward function parameters might require grid searching, and the RL training process is computationally expensive. Future work could explore more dynamic knowledge boundary learning methods, investigate applicability across a broader range of tasks, and aim to reduce training resource requirements.

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

## A    LLM USAGE DISCLOSURE

We use LLM for paper writing to check grammar and boost the clarity. We do not use LLM for research. We do not use LLM to generate experiment code and analysis.

## B    IKEA AGENT TEMPLATE

We use the system template in Table 5 to prompt the agent to interact with the environment:

## C    DATASET CONSTRUCTION

We use NQ (Kwiatkowski et al., 2019) and HotpotQA (Ho et al., 2020) as the in-distribution datasets. We use the PopQA (Mallen et al., 2023) and 2Wikimultihopqa (Ho et al., 2020) as the out-of-distribution datasets. Following the knowledge-boundary training dataset construction method, we construct easy and hard subset for each dataset. We use the Qwen-2.5-3B-Instruct as the sampling model. There are 512 examples in each subset of each dataset.

## D    BASELINES

We compared methods that do not require training (e.g., zero-shot or few-shot prompting), those that utilize Supervised Fine-Tuning (SFT) and Direct Preference Optimization (DPO), and reinforcement learning-based approaches. The corresponding baselines are shown as follows:

- **Direct** We directly prompt the model to answer the relevant question using only its internal knowledge.

- **RAG** We retrieve documents using the question and prompt the model to answer the relevant question relying solely on the retrieved knowledge.

- **Iter-Retgen** (Shao et al., 2023) It is an iterative retrieval-generation method that achieves strong performance by synergizing parametric and non-parametric knowledge. We set the default ret-gen turns as 4.

- **IR-COT** (Trivedi et al., 2023) It is method for multi-step question answering, which interleaves retrieval with steps in the chain-of-thought, using CoT to guide retrieval and retrieval results to improve CoT. It will adatpively determine the turns of retrieval according to the knowledge needs. And we set the max turns as 4.

- **FLARE** (Jiang et al., 2023) This method introduces a forward-looking active retrieval-augmented generation (FLARE) approach that iteratively uses predictions of upcoming sentences to anticipate future content and retrieves relevant documents when a sentence contains low-confidence tokens, in order to regenerate that sentence. It uses a specific criteria to determine the retrieval timing. We set the max number of search as 4.

- **DeepRAG** (Guan et al., 2025) This method introduces a framework that models retrieval-augmented generation as a Markov Decision Process (MDP), enabling strategic and adaptive retrieval to improve retrieval efficiency and answer accuracy. It collects offline trajectories to finetune the base model with SFT and DPO.

- **R1** (DeepSeek-AI et al., 2025) It uses reinforcement learning to encourage the model to reason in order to activate its internal knowledge. This method only uses the internal knowledge.

- **Search-R1** (Jin et al., 2025; Song et al., 2025; Chen et al., 2025) The model's capacity to employ external retrieval tools is activated via multi-turn reinforcement learning. This technique exclusively relies on the model's external knowledge. We set the max number of search as 4.

You are an expert assistant capable of solving knowledge-intensive tasks efficiently. You will be given a question to answer as accurately as possible.

You can use your own knowledge or call external search engines to gather additional information, but searching should only occur when necessary. Specifically, you should search only when encountering a clear knowledge gap or uncertainty that prevents you from confidently answering the question.

To arrive at the answer, you will proceed step-by-step in a structured cycle of '<think>thinking content</think>', '<search>search query</search>' (optional), and '<context>returned external information</context>' (optional) sequences. You can only generate content within these special tags.

Remember that <search>xxx</search> and <context>xxx</context> are optional. You can skip them if you have enough knowledge to answer the question. And skip is them is encouraged and preferable.

Thinking Phase (<think>): Recall your own knowledge, analyze current information, and decide whether further search is needed. If enough knowledge is available, skip searching. For question, it may be decomposed into sub-questions for you to think about. Some sub-questions may be answered by searching, while others may not. You can also use the <think> tag to express your uncertainty about the sub-question.

Searching Phase (<search>): Formulate a search query only if required to fill a knowledge gap or verify uncertainty. Skip if unnecessary. Information Phase (<context>): Use search results as context for further steps. If no search was performed, proceed without this phase.

Answering Phase (<answer>): Provide a concise and accurate answer within <answer> tags once you have enough knowledge. The answer should be short and precise, such as <answer> Beijing </answer>.

Here are a few examples:

—

Example 1: search is needed, search more than once
Question: xxx
<think> xxx </think>
search> xxx </search>
<context> xxx </context>
<think> xxx </think>
(search more than once)
<think> xxx </think>
<answer> xx </answer>
Example 2: search is needed, only search once
Question: xxx?
<think> xxx </think>
<search> xxx </search>
<context> xxx </context>
<think> xxx </think>
<answer> xxx </answer>

—

Example 3: search is not needed
Question: xxx?
<think> xxx </think>
<answer> xxx </answer>

—

You can search 0 - N times. 0 is preferable. Each search should be focused on one sub-question. The answer within <answer> tags should be short and precise, such as <answer> yes </answer>.
Now it is your turn to answer the question.
Question: {question}

Table 5: System prompt of IKEA.

## E  IMPLEMENTATION DETAILS

We use e5-base (Wang et al., 2022) as the retriever model and wikipedia2018 as the corpus for retrieval. We employ Qwen2.5-3B(-Instruct) and Qwen2.5-7B(-Instruct) as the initial models. Models with the "-Zero" suffix are trained from the Base model, while those without it are trained from the Instruct model. we use FlashRAG (Jin et al., 2024) to reproduce the baseline results. We utilize the verl (Sheng et al., 2025) framework for training. GRPO (Shao et al., 2024) is used as the reinforcement learning algorithm. We use the NQ and HotpotQA to construct training datasets. For each one, we sample 4000 easy samples and 4000 hard samples. We set the number of rollouts as 16 for one task. We set the learning rate as 5e-7, warmup ratio as 75%, batch size as 256, training steps as 120. We set $r_{kb+}$ as 0.6 and $r_{kb-}$ as 0.05, $RT_{max}$ as 3. We use 8 A100 GPUs for all the experiments.

## F  CASE STUDIES

**Case Study 1: Avoiding Invalid Retrieval**

**Question:** Identify the American comedian born on March 21, 1962, who appeared in the movie "Sleepless in Seattle."
**Ground Truth:** Rosie O'Donnell

---

**Trajectories** (some content omitted due to context limit)

**R1**: <think> Let's think through this step by step: 1. The question asks about an American comedian born on March 21, 1962, who appeared in the movie Šleepless in Seattle.2. The movie Šleepless in Seattlewas released in 1993. 3. Some of the main cast members of the movie include Tom Hanks, Meg Ryan, and Sally Field. 4. Among these actors, the only comedian who was active in the 1990s and matches the birth year of 1962 is Billy Crystal. 5. Billy Crystal is an American comedian who was indeed active in the 1990s and matches the birth date given. </think> <answer> Billy Crystal </answer>

**Search-R1**:
<think> To answer this question, I need to identify an American comedian born on March 21, 1962, who appeared in the movie Šleepless in Seattle.Ï don't immediately recall this specific information, so I will need to search for it. </think>
<search> American comedian born on March 21, 1962, appeared in Sleepless in Seattle </search>
<context>Doc 1(Title: Šleepless in Seattle) (content omitted) Doc 2(Title: M̈ike Birbiglia) (content omitted) Doc 3(Title: D̈ave Attell) (content omitted) </context>

... (the last two turn are exact the same as this turn)

**IKEA**:
<think> I need to identify the American comedian born on March 21, 1962, who appeared in the movie Šleepless in Seattle.This requires specific information about the cast of the movie and the birth date of the comedian. </think>
<search> American comedian born on March 21, 1962, appeared in the movie Šleepless in Seattle</search>
<context>Doc 1(Title: Šleepless in Seattle) (content omitted)Doc 2(Title: R̈oss Malinger) (content omitted)Doc 3(Title: R̈oss Malinger) (content omitted)</context>
<think> Ross Malinger was born on July 7, 1984, not March 21, 1962. The correct comedian is Billy Crystal, but Billy Crystal was not born on March 21, 1962. The correct comedian is actually Rosie O'Donnell, who was born on March 21, 1962, and appeared in the movie Šleepless in Seattle.</think>
<answer> Rosie O'Donnell </answer>

---

**Analysis**

**R1's Behavior (Incorrect):**

- The model incorrectly identifies Billy Crystal as the answer, resulting in a factual hallucination.

**Search-R1's Behavior (Incorrect):**
- The model fails to find the correct answer even after multiple searches. The first search was effective and contained the entity "Rosie O'Donnell" but not her birthday, which prompted further searches.
- Subsequently, the model failed to decompose the task and repeatedly generated the same ineffective query. The reasoning process terminated after hitting the search limit, leading to its inability to provide the correct answer.

**IKEA's Behavior (Correct):**
- IKEA generates a search query and ultimately identifies the correct answer as Rosie O'Donnell. The first search contains the entity "Rosie O'Donnell" but not her birthday.
- In its subsequent reasoning process, it first eliminates Ross Malinger and Billy Crystal for not matching the birth date. It then confirms that Rosie O'Donnell is the American comedian born on March 21, 1962, who appeared in "Sleepless in Seattle," satisfying all conditions. This demonstrates a classic example of synergistic reasoning between internal and external knowledge.

---

**Case Study 2: Avoiding Conflict Retrieval**

**Question:** Identify the founder of the team Brad Budde played for, which was founded in 1960 as the Dallas Texans.
**Ground Truth:** Lamar Hunt

---

**Trajectories** (some content omitted due to context limit)

**R1**: <think> Let's think through this step by step: 1. Identify the team: Brad Budde played for the Kansas City Chiefs. The Kansas City Chiefs were previously known as the Dallas Texans. 2. Determine the founding year: The Dallas Texans were founded in 1960. 3. Identify the founder: The Dallas Texans were founded by Lamar Hunt. </think> <answer> Lamar Hunt </answer>",

**Search-R1**: <think> I need to find out who founded the Dallas Texans in 1960. </think>
<search> who founded the Dallas Texans in 1960 </search>
<context>Doc 1(Title: D̈allas Texans (NFL)) (content omitted) Doc 2(Title: T̈ex Schramm) (content omitted) Doc 3(Title: Ḧistory of the Dallas Cowboys) (content omitted)</context>
From the information provided, it seems Lamar Hunt founded the Dallas Texans in 1960 as part of the American Football League (AFL). However, the Dallas Cowboys were established in the same year as an NFL team. The key detail I need to confirm is the specific person who founded the Dallas Cowboys in 1960, which is Tex Schramm.
<search> Tex Schramm founded Dallas Cowboys </search>
<context>Doc 1(Title: T̈ex Schramm) (content omitted) Doc 2(Title: T̈ex Schramm) (content omitted) Doc 3(Title: T̈ex Schramm) (content omitted)
<answer> Tex Schramm </answer>

**IKEA**: <think> I need to identify the team that Brad Budde played for, which was founded in 1960 as the Dallas Texans. Then I need to find out who founded this team. </think>
<search> Brad Budde NFL team founded 1960 Dallas Texans </search>
<context>Doc 1(Title: 1̈960 Dallas Texans season) (content omitted) Doc 2(Title: D̈allas Texans (NFL)) (content omitted) Doc 3(Title: K̈ansas City Chiefs) (content omitted) </context>
<think> Based on the information gathered, the Dallas Texans were founded in 1960 by Lamar Hunt. </think>
<answer> Lamar Hunt </answer>",

---

**Analysis**

**R1's Behavior (Correct):**
- The model's reasoning process is clear and correct. It first identifies that Brad Budde played for the Kansas City Chiefs, whose predecessor was the Dallas Texans, founded in 1960. It correctly concludes that the founder was Lamar Hunt.

**Search-R1's Behavior (Incorrect):**
- The model's error lies in confusing the Dallas Texans with the Dallas Cowboys.
- Although it retrieved some information about the Dallas Cowboys and their related figure, Tex Schramm, this was irrelevant to the question. Search-R1 retrieved seemingly relevant but actually unrelated information, which misled the model's reasoning and overrode its originally correct parametric knowledge.

**IKEA's Behavior (Correct):**
- The model's solution is correct. Through a better search query, it clearly identified that the predecessor of Brad Budde's team was the Dallas Texans, founded in 1960 by Lamar Hunt.
- The reasoning is accurate, and the answer is correct. By using a better search query, IKEA retrieved more relevant information and avoided the interference of conflicting information.

