# OpenReview forum: "Reinforced Internal-External Knowledge Synergistic Reasoning for Efficient Adaptive Search Agent"
_ICLR.cc/2026/Conference — ICLR 2026 Conference Withdrawn Submission_

### Official Review · Reviewer_mATT · 2025-10-24

**Soundness:** 2
**Presentation:** 3
**Contribution:** 2
**Rating:** 2
**Confidence:** 4

**Summary:**

This paper addresses issues in existing RAG and RL-based LLM search agents—such as underused internal knowledge, redundant retrievals, knowledge conflicts, and high latency—by proposing IKEA, a reinforced reasoning agent.
IKEA identifies its own knowledge boundaries, prioritizes internal knowledge, and uses external search only when internal knowledge is insufficient. It achieves this via a novel knowledge-boundary aware reward function and training dataset, which guide RL to balance accuracy, minimal unnecessary retrievals, and appropriate external searches.
Evaluations show IKEA outperforms baselines on multiple knowledge reasoning tasks, reduces retrieval frequency, and has strong generalization.

**Strengths:**

Addresses a practical issue in Retrieval-Augmented Generation (RAG) systems: over-reliance on retrieval even when internal knowledge is sufficient, which leads to increased latency and potential knowledge conflicts.
The design of knowledge boundary-aware rewards is intuitive and reasonable, with clear hierarchical structure for behavioral preferences.
Evaluated on multiple datasets, including both in-distribution and out-of-distribution tests.

**Weaknesses:**

1. Limited technical innovation. Its reward shaping strategy for existing Reinforcement Learning (RL) methods essentially only adopts the "Search-R1" framework combined with reward function construction.
2. The comparison scope for baseline model construction is limited and not fully expanded.
3. The 1:1 ratio of easy-to-hard questions lacks theoretical basis, and no ablation experiments have been conducted under different ratios.
4. Lacks a formal and rigorous definition of the "knowledge boundary".
5. The classification of easy questions (Qeasy) and hard questions (Qhard) is overly simplified. The criterion of "at least one correct answer in N samplings" ignores the confidence gradient, and no tests have been conducted on multi-hop reasoning tasks with ambiguous knowledge boundaries.

**Questions:**

1. Beyond in-context learning, what would be the effect of using other knowledge probing techniques?
2. Conduct more comprehensive comparative experiments with baseline models.
3. Evaluate performance on a wider range of tasks beyond question answering.

---

### Official Review · Reviewer_u7wK · 2025-10-31

**Soundness:** 3
**Presentation:** 3
**Contribution:** 3
**Rating:** 6
**Confidence:** 4

**Summary:**

This paper proposes the reinforcement learning framework IKEA, which enables large language models to adaptively balance the use of internal (parametric) knowledge and external (retrieval-based) knowledge.
The core idea is to train the model to recognize its knowledge boundary—to reduce retrieval when internal knowledge is sufficient and to trigger retrieval when it is not.
The authors design:
- a knowledge-boundary-aware reward function that encourages the model to obtain correct answers with minimal retrieval, and
- a mixed-difficulty training set that helps the model learn a dynamic balance between internal and external knowledge usage during reinforcement learning.

On datasets including NQ, PopQA, HotpotQA, and 2Wiki, IKEA maintains or improves accuracy while reducing the average number of retrievals by 35–50% compared to Search-R1.

**Strengths:**

1. Proposes a reinforcement-learning-based framework for internal–external knowledge synergistic reasoning.
2. The reward function is well-designed, explicitly modeling the trade-off between retrieval cost and answer accuracy.
3. Achieves stable performance improvements across multiple datasets.

**Weaknesses:**

1. Limited generalization on Hard/OOD scenarios: Although IKEA aims to trigger retrieval when internal knowledge is insufficient, its improvements over Search-R1 on Hard/OOD datasets (e.g., PopQA and 2Wiki) are modest or even negative, suggesting that the model still tends to over-rely on internal knowledge in out-of-distribution settings.
2. Mismatch between model scale and training data: Models of different sizes have distinct distributions of internal parametric knowledge.
If the same training data are used for all model scales, the “knowledge boundaries” learned by IKEA may not reflect the actual knowledge coverage of each model. It is recommended that the training data be separately sampled or adaptively constructed for different model sizes to ensure that boundary learning aligns with each model’s internal knowledge structure.

**Questions:**

1. Do the Qwen-2.5-3B and 7B models use the same training data? How does model scale affect boundary-learning behavior?
2. What are the main factors behind IKEA’s limited improvements on Hard/OOD tasks compared to Search-R1?

---

### Official Review · Reviewer_jjGF · 2025-11-01

**Soundness:** 2
**Presentation:** 3
**Contribution:** 2
**Rating:** 4
**Confidence:** 3

**Summary:**

This paper proposes the Reinforced Internal-External Knowledge Synergistic Reasoning Agent (IKEA), which aims to optimize the reasoning capabilities of large-scale language models in knowledge-intensive tasks through reinforcement learning. The core innovation of IKEA lies in enabling the model to dynamically perceive its knowledge boundary, allowing it to intelligently decide when to rely on internal knowledge and when to perform external retrieval. By designing a refined reward mechanism, IKEA reduces redundant retrieval operations, maximizes the efficiency of internal knowledge usage, and ensures the effective supplementation of external knowledge.

**Strengths:**

1. Building upon previous methods such as Search-R1, this paper introduces an exploration of the balance between internal and external knowledge paths. This not only empowers the model with the ability to actively explore retrieval but also enables it to learn the boundary between its internal parametric knowledge and external knowledge.
2. The baseline experimental comparison is thorough, and the writing is clear.

**Weaknesses:**

1. The novelty of this paper is limited. It appears to merely integrate the exploration path strategy for balancing internal and external knowledge from DeepRAG into the Search-R1 framework. Essentially, it only designs a more complex reward function based on Search-R1, incorporating four different key behaviors, and is essentially an extension of DeepSearch-like works.
2. The paper lacks appropriate experimental design to support the claim that its method is indeed more effective in learning the boundary between internal and external knowledge.

**Questions:**

1. Does the baseline method, such as Search-R1 or DeepRAG, used for training align with the method proposed in this paper on the training data?

---

### Official Review · Reviewer_KFrF · 2025-11-04

**Soundness:** 2
**Presentation:** 2
**Contribution:** 2
**Rating:** 6
**Confidence:** 2

**Summary:**

IKEA introduces (i) a knowledge-boundary–aware reward that pays more for correct answers with fewer searches and gently penalizes fruitless searches on wrong answers, and (ii) a balanced training set that mixes questions the model can already answer (Qeasy) with ones it likely cannot (Qhard). Together these push the policy to rely on internal knowledge first and retrieve only when necessary.

**Strengths:**

- Directly optimizes “when to search.” The reward prefers “correct with fewer RT” over “correct with more RT,” and prefers “searched but wrong” over “did not search and wrong,” which encodes a clear ordering of behaviors (1>3>4>2 in the paper’s terms). This is a simple, targeted shaping for the retrieval timing problem rather than a classifier-gated or imitation-heavy approach.


- Balanced difficulty splits (Qeasy/Qhard) are used not just for evaluation but as a training prior to keep the policy from drifting to extreme “always/never retrieve” strategies. The ablations show this matters.


- This is not a new tool interface or planner; the novelty is in reward design + data curriculum that together teach knowledge-boundary awareness inside standard RLVR.

**Weaknesses:**

- Hyperparameter sensitivity. The balance between accuracy and search frequency depends on RTmax, rkb+, and rkb−. Without concrete values and sweeps, it is hard to know how portable the setting is across model sizes and retrievers.


- Labeling procedure. The Qeasy/Qhard split is defined by the base model’s own probe with N samples. This can drift with model choice and N; it would be helpful to report label distributions and sensitivity to N.


- Corpus constraint. Experiments fix Wikipedia2018 and e5-base. Showing results with a web search API or a noisier corpus would better test robustness to non-curated text and long-tail entities.


- Latency claim by proxy. The paper argues fewer searches mean lower latency, which is usually true, but wall-clock time is not reported. Reporting actual runtime improvements (including network/IO) would strengthen the efficiency claim.


- Metric scope. EM is a strict measure. Including F1 or calibration/abstention metrics would help assess whether “no search” answers degrade gracefully.

**Questions:**

See Weakness

---

### Official Review · Reviewer_P6Qj · 2025-11-05

**Soundness:** 2
**Presentation:** 3
**Contribution:** 2
**Rating:** 4
**Confidence:** 4

**Summary:**

This paper proposes a reinforcement learning framework that aims to train an adaptive search agent capable of balancing internal parametric knowledge and external retrieval when answering queries. The model is rewarded for preferring internal reasoning when sufficient and only invoking search when necessary, with the goal of improving both accuracy and efficiency. Experiments on several RAG-style QA datasets show reduced search calls and improved correctness compared to search-heavy baselines.

**Strengths:**

1. The paper addresses an important challenge in RAG systems: when to retrieve rather than how to retrieve.
2. The reward formulation is intuitive and well-motivated, in that it attempts to explicitly model “knowledge boundary.”
3. Results demonstrate that the proposed framework can reduce retrieval calls while maintaining or improving answer correctness.
4. The paper is generally well-written and follows a clear structure.

**Weaknesses:**

1.	The data construction pipeline (line 245) suggests a much simpler alternative:
Train a difficulty classifier to decide whether a question requires retrieval.
This is directly in line with Adaptive-RAG: Learning to Adapt Retrieval-Augmented LLMs through Question Complexity.
Such baselines are not included, leaving it unclear whether RL is actually necessary — or whether the same behavior can be achieved more simply and without retraining the model.
In particular, for practical deployment (e.g., Qwen DeepSearch-Agent), RL-based retraining introduces substantial cost, while a lightweight adaptive controller would not.

	2.	The experiments focus on datasets where retrieval depth and search horizon are relatively small.
To convincingly demonstrate the value of “adaptive search,” evaluation should include more complex multi-step environments, such as BrowserGym, WebArena, BC, or GAIA, where search depth and branching are significantly larger.
Without these evaluations, it remains unclear whether the proposed method provides benefits in realistic, long-horizon reasoning tasks.

3.	Unclear whether the model truly learns knowledge boundaries: The performance improvements may arise simply from data distribution differences, rather than from the model actually understanding when internal knowledge is sufficient.
The paper does not provide diagnostics showing that the model has learned a meaningful internal-external knowledge separation.
Suggested analyses that are currently missing include:
	•	Cross-distribution generalization tests
	•	Behavioral probes to evaluate reliance on parametric knowledge
	•	Visualization of search decision patterns independent of dataset structure
Without these, the claimed “knowledge boundary learning” remains unsubstantiated.

**Questions:**

1.	Can the authors compare against a classifier-based gating approach that selects between direct answering and Search-R1-style reasoning?
	2.	How does the proposed method perform in long-horizon web/agent environments, where search planning is necessary beyond RAG retrieval?
	3.	Can the authors provide behavioral evidence that the model learned knowledge boundaries rather than dataset heuristics?
	4.	Is it possible to design an adaptive deepsearch mechanism that does not require RL training?

---

### Note · Authors · 2026-01-04

I have read and agree with the venue's withdrawal policy on behalf of myself and my co-authors.